# The Impact of Physical Properties on the Leaching of Potentially Toxic Elements from Antimony Ore Processing Wastes

**DOI:** 10.3390/ijerph16132355

**Published:** 2019-07-03

**Authors:** Saijun Zhou, Andrew Hursthouse

**Affiliations:** 1College of Civil Engineering, Hunan University of Science and Technology, Xiangtan 411201, China; 2Hunan Provincial Key Laboratory of Shale Gas Resource Utilization, Hunan University of Science and Technology, Xiangtan 411201, China; 3School Computing, Engineering & Physical Science, University of the West of Scotland, Paisley PA1 2BE, UK

**Keywords:** antimony ore processing wastes, particle size, leaching, potentially toxic elements

## Abstract

This study reports on the assessment of the impact of antimony mine wastes from Xikuangshan (XKS) Antimony Mine in Lengshuijiang City, Hunan Province. We focus on the leaching of a number of potentially toxic elements (PTEs) from residues from the processing of antimony ore. The PTE content of ore processing waste and solutions generated by leaching experiments were determined for a suite of PTEs associated with the ore mineralization. These were Sb, As, Hg, Pb, Cd and Zn. As anticipated, high concentrations of the PTEs were identified in the waste materials, far exceeding the standard background values for soil in Hunan Province. For Sb and As, values reached >1800 mg·kg^−1^ and >1200 mg·kg^−1^, respectively (>600 and >90 times higher than the soil background). The leaching of Sb, As, Hg, Pb, Cd and Zn decreased with an increase in grain size and leachable portions of metal ranged between 0.01% to 1.56% of total PTE content. Leaching tests identified the release of PTEs through three stages: a. alkaline mineral dissolution and H^+^ exchanging with base cation; b. oxidation and acid production from pyrite and other reducing minerals; and c. the adsorption and precipitation of PTEs.

## 1. Introduction

The exploitation of antimony by mining has created serious environmental pollution at a number of locations and has attracted international research interest in understanding the mechanisms driving impact [1,2,3,4]. The waste produced by mining not only occupies large areas of land, but also causes lasting environmental pollution, particularly by the harmful elements released after heavy rainfall [4,5,6,7,8]. China has the most abundant Sb resources in the world [9] and has been responsible for approximately 90% of all antimony metal produced globally over the past decades. The Xikuangshan (XKS) Mine in Hunan, Central South China is the world’s largest Sb mine, and has been reported to produce 25% of the world’s Sb. It has been in operation extracting and smelting for more than 120 years, which coupled with the large-scale mining operations has produced significant residual materials open to the percolation of water. It has released significant quantities of harmful elements like Sb, As, Hg, Pb, Cd and Zn to the surroundings, threatening the wider environment and human health [10,11,12,13]. When an excessive amount of Sb enters the human body, it causes diseases to the liver, skin, and respiratory and cardiovascular systems, or even cancer [14,15]. The element Sb and its compounds are listed as pollutants of priority interest by the United States Environment Protection Agency [16] and the European Union [17,18].

Up to now, studies on pollution form antimony ore extraction and processing are mainly focused on Sb in waste water, waste gas and waste solid residues from antimony beneficiation and metallurgy, the distribution and spatial characteristics of pollution, the factors affecting its migration and release, transformation of chemical forms and its biogeochemical behavior, etc. [19,20,21,22,23]. Few studies exist on the pollution caused from effects of the environment on waste rock dumps. A number of studies showed that the dissolution of Sb-bearing ore minerals is thought to contribute significantly to pollution in the local drainage zone. The kinetics of the mobilization of Sb from stibnite, Sb_3_O_6_OH and Sb_2_O_3_ under environmental conditions has been studied in detail by Biver and Shotyk [7,8]. Hu et al. [24,25] studied the release kinetics and mechanisms of Sb from stibnite and Sb_2_O_3_ under the irradiation of light. Guo et al. [5] studied the leaching characteristics of antimony smelting slag in XKS Sb Mine and found that smelting slag is an important source of Sb pollution in nearby farmland. This paper reports on the effect of environmental conditions on the ore processing waste rock stored at a the XKS Sb Mine in Hunan Province, Central South China. It assesses leaching processes affecting waste rock of different particle sizes, identifying changes in the leaching solution (e.g., pH, conductivity and PTE release). The impact of results on prediction of wider environmental and ecological risk are also addressed.

## 2. Material and Methods

### 2.1. Experimental Material

We collected the Sb ore waste rocks for our experiments from an area of open waste storage near the northern part of the XKS Sb mine (E 111°29′34.87″–E 111°29′43.45″, N 27°46′06.70″–N 27°46′13.41″) in Lengshuijiang City, Hunan Province. The waste rock storage was over an area of 36,000 m^2^, with a total stockpile of about 850,000 tons. A series of 20 sampling sites (see Figure 1) were selected during site walk. At each point samples were collected from within a square (1 m × 1 m), with a composite sample of one kilogram slag material collected at the center and its four equidistant points of the sampling site. The 20 samples were mixed uniformly, dried naturally, and then ground in a mill (XMQФ240×90, Jiangxi Ganye, China) to three grades [26] with particle sizes 0.075–0.15 mm (1), 0.15–0.425 mm (2) and 0.425–0.85 mm (3), for experimentation. The particle size distributions of antimony ore processing wastes are shown in Figure 2.

### 2.2. Test Instruments and Reagents

The instrumental techniques used in sample analysis were: X-ray diffraction (XRD, D8 Advance, Bruker AXS Germany), hydride generation-atomic fluorescence (AFS-9700, Beijing Haiguang, China), oven heating (GZX-9246MBE, Shanghai Boxun), scanning electron microscopy (SEM, JSM-6380LV, JEOL, Japan), energy-dispersive spectrometry (EDS, GENESIS, EDAX, United States), pH metering (PB-10, Sartorius, Germany), conductivity metering (CCT-5320E, Hebei Ruida, China), use of an ultra-pure water machine (LBY-20, Chongqing Owen Technology Co., Ltd., Chongqing, China), and orbital shaking (HY-5, Jiangsu Jintan, China). The reagents used (HNO_3_, HF and H_2_O_2_) were of high purity (China Chemicals Co., Ltd., Shanghai, China). All utensils are soaked in 10% nitric acid solution for 48 h before use, and then rinsed with ultra-pure water.

### 2.3. Basic Mineral Composition of Sb Ore Waste Rocks and PTE Determination

The mineral composition of antimony ore waste rocks was observed and analyzed by XRD. The PTEs were determined by AFS (Hydride Generation-atomic Fluorescence) after digestion. The digestion process involved: Five samples of particle size 1 (0.100 g) were placed in PTFE digestion vessels along with one blank sample; acid was added (5 mL of HNO_3_ and 0.5 mL HF), and allowed to digest at 170 ℃ for 12 h. After cooling, 1 mL of 30% H_2_O_2_ was added and allowed to react for 30 min. A 10 mL aliquot of 5% nitric acid solution was then added and filtered by with a polyethylene film injection filter (0.2 μm). The filtrate was then diluted to 50 mL with ultra-high purity water and kept at 4 °C before analysis by AFS.

### 2.4. Experimental Methods

The leaching solution was a mixture of H_2_SO_4_ and HNO_3_ at the volumetric ratio of 3:1, diluted with NaOH and high purity water, and its initial pH value is set at 4.98, according to the rainfall of Lengshuijiang City [27]. Firstly, 25 g waste rock samples with different particle sizes were loaded into a 1000 mL glass bottle with stopper grinding mouth respectively, and then the leaching solution (250 mL) was added into the bottle. The solid-liquid ratio was set at 1:10. The glass bottles with stopper were placed in a circular shaker with rotational speed of 120 ± 5 rpm for the leaching test.

The leaching test lasted 12 days. An aliquot of 10 mL of leachate was taken at regular intervals every day to measure its pH value and conductivity; and the concentration of PTEs (Sb, As, Hg, Pb, Cd, Zn) was determined for each sample of leachate. The leachate consumed during the test was replaced by blank leaching solution to maintain the solid to liquid ratio. Three replicates were run in parallel. The concentration of PTEs in leachate was determined by AFS-9700, and pH and conductivity measured. After leaching solid samples were analyzed by scanning electron microscopy-energy dispersive spectrometry (SEM-EDS), and element composition was analyzed. The dissolution of PTEs per unit mass is calculated by the following formula:(1)Q=q·v
(2)R=(Q/c)×100%

In the formula, *Q* is the amount per unit mass of the PTEs dissolved (μg·kg^−1^); *q* is the concentration of the PTEs in the leachate (μg·L^−1^); *v* is the volume of the leachate (L); *R* is the amount of dissolution for each PTE; *c* is the concentration of PTE per unit mass of raw solid (μg/kg).

### 2.5. Quality Control

To ensure the data accuracy in the analysis process and the stability of the testing instruments, the standard reference soil (GBW07406) from the National Standard Material Center was digested and measured by the same method as the antimony ore waste rocks samples. The recovery rates of Sb, As and Hg in standard reference materials are 95–106%, 94–107% and 94–104%, respectively. At the same time, in each batch of analytical samples, reagent blanks were added, and 20% of the samples were re-measured. The relative standard deviation (RSD) of re-measurement was less than 10%.

## 3. Results and Discussion

### 3.1. Basic Mineral Composition and Heavy Metal Content of Sb Ore Waste Rocks

The XRD results for samples of waste rocks are shown in Figure 3. The results show that the main minerals identified were quartz and calcite, and the samples also contained a small amount of stibnite and pyrite. As shown in Table 1, the content of Sb and As in waste rocks was very high, reaching 1806.21 mg·kg^−1^ and 1277.64 mg·kg^−1^, respectively, which is 606 and 91 times higher than the soil background values for PTEs in Hunan Province [28]. Although the Hg and Cd contents were relatively low (1.44 mg·kg^−1^, 1.67mg·kg^−1^), they were still 16 and 21 times higher than the standard value. Those are potentially hazardous materials.

### 3.2. Leaching Law of Antimony Ore Waste Rocks with Different Particle Size

The change in leachate pH value is an important indicator for evaluating environmental pollution from mining, and it also has a strong influence on the valence and fractionation of PTEs in samples [29]. Conductivity can reflect the concentration of PTEs in leachate, and its trend can reflect the intensity of interaction between solid-liquid interface of leaching system. The results for experimentation carried out are shown in Figure 4, Figure 5, Figure 6, respectively.

As shown in Figure 4, the trend in pH of the leachate with different particle sizes was essentially the same. It can be divided into three stages: an initial increase to a maximum, a slow decline and stabilization. On the 6th day of leaching, the pH of the leachate for particle size 1 and particle size 2 increased rapidly to the highest value, with particle size 1 = pH 8.47, while for particle size 3 the highest value of 7.98 was achieved on the 7th day. After that, the pH values began to decrease, with the leachate from particle size 1 and 2 dropping to their lowest points on the 9th day (particle size 2 was pH 7.94), 0.21 lower than that of the leachate on the 6th day. Between the 10th day and the 12th day, the pH values gradually stabilized, and the final leaching solutions were alkaline.

From Figure 5, it can be seen that the conductivity of the leachate decreased with the increase of particle size. Over the first eight days, the leachate conductivity increased rapidly, but slowed and stabilized between Day 9 to Day 12. On the 12th day, the conductivity for all three wastes was nearly 400 μs·cm^−1^.

As can be seen from Figure 6, the leaching behavior for all six PTEs was variable from element to element. It is clear that the portion solubilized decreases with the increase of particle size, following the trend in surface area available to react. After 12 days of leaching, the cumulative amount leached for the six PTEs (calculated according to Formula 1 and calculated in 10 mL leaching solution of each sample) was in the order Sb > Zn > Pb > As > Hg > Cd. Figure 6a–f shows that the leaching amount of six heavy metals basically reached a stable value in 9–12 days. After 12 days of leaching, the total amounts leached from the waste are shown in Table 2. The leaching characteristics of heavy metals in antimony ore processing wastes are consistent with previous studies [26,30].

It can also be seen from Table 2 that the leaching of Sb and Zn from the waste rock with different particle sizes differed greatly. The leaching of Sb was far below 18% [4]; the reason may be related to the different pH of leaching and associated solubility. The smaller the particle size, the higher the leaching of the PTEs, while the leaching of As, Hg, Pb and Cd was less affected by particle size.

### 3.3. Scanning Electron Microscope and Energy Spectrum Analysis

SEM and EDS analysis were used to study antimony ore waste rock (0.075–0.15 mm) before and after leaching, and the results of the EDS measurement of waste rock surface morphology at different stages were obtained. Figure 7 shows the surface morphology of the waste before and after leaching. It can be seen that the surface morphology changed significantly. Before leaching, the surface was relatively flat, compact and less crystalline, while after leaching, the surface of the waste was broken, with a small amount of granular sediment.

Table 3 shows EDS results for the waste before and after leaching, emphasizing the very significant changes in the surface composition. Before leaching, the main elements on the surface of waste rock were O, Si and Al. The contents of target PTEs Sb, As, Hg, Pb, Cd and Zn in the waste were 1.42%, 0, 0.07%, 1.09%, 0.12% and 1.24%, respectively. With the increase of leaching time, the content of Si on the surface of waste antimony ore rock decreased; that is, the content of Si-bearing quartz or silicate minerals on the surface of waste rock decreased. The contents of PTE elements and oxygen increased and varied greatly, which indicates that the PTE on the surface of and inside the waste diffuse into solution under erosion, and then precipitate with metal hydroxides and as complexes [31,32], and are adsorbed on the surface of waste rock. The increase of As content may have been due to the exposure of As encapsulated in the waste rock particles of antimony ore rock by acid leaching.

### 3.4. Leaching Mechanism of Antimony Ore Waste Rock

During the leaching process, the leachate pH showed three distinct stages: An initial increase, decrease and stabilization. The stages also reflected differences in the chemical stability. When the pH rose (the first 6 days of leaching), the leachate conductivity increased rapidly, with increased H^+^ consumption by reactions—probably solubilizing minerals such a as calcite, with H^+^ in solution interacting with available cations (K^+^, Na^+^, Ca^2+^, Mg^2+^, Al^3+^) [33] (Equation (3)), resulting in the rapid increases in pH and conductivity.
(3)$$R^{n-}X^{n+}+nH^{+}↔R^{n-}H^{n+}+X^{n+}\;X=K/Na/Ca/Mg/Al$$

Most of the metal elements in antimony ore waste rock predominantly exist in a very stable residual fraction and organic-sulfide fraction. Other phases include the reducible fraction of iron and manganese oxide, the carbonate-bound fraction and exchangeable fraction [34]. Metal elements in carbonate-bound and exchangeable forms are easily soluble in water and acid solution [35]. The Sb associated with Carbonate-bound and exchangeable forms in the waste was released by Equations (4)–(9) below [4,36]. Sulfur species in the dissolution system also governed the dissolution equilibrium of Sb_2_S_3_ [37]. The oxidation of sulfides shifted the dissolution equilibrium of Sb_2_S_3_. The released sulfide from upon dissolution of stibnite can undergo oxidation quickly, as observed during previous experiments [7]. The associated As, Hg, Pb, Cd and Zn in the leachate were also due to the solubility of carbonate-bound and exchangeable forms in acid.
(4)$$Sb_{2}S_{3}(s)+6H_{2}O(l)↔2Sb{\left(OH\right)}_{3}(aq)+3H_{2}S(aq)$$
(5)$$Sb{\left(OH\right)}_{3}^{0}+1/2O_{2}+2H_{2}O\to 2Sb{\left(OH\right)}_{6}^{-}+H^{+}$$
(6)$$H_{2}S+2O_{2}\to H_{2}SO_{4}$$
(7)$$2HS^{-}+2O_{2}\to H_{2}O\;+S_{2}O_{3}^{2-}$$
(8)$$S_{2}O_{3}^{2-}+4O_{2}\to 2SO_{4}^{2-}+2S^{0}$$
(9)$$2HS^{-}+4O_{2}\to 2SO_{4}^{2-}+2H^{+}$$

During the decline in pH (7th–10th day), the conductivity of leachate continued to increase, but the rate of increase slowed, indicating that the exchange between H^+^ dissolution and basic cations was still strong at this time, but the intensity was decreasing. As leaching progressed, the portion of reducing minerals in the waste increased, and pyrite was oxidized (Equation (10)), producing H^+^ to reduce leachate pH, which conforms to the law of leachate pH reduction in the leaching system.
(10)$$FeS_{2}+15O_{2}+2H_{2}O↔4Fe^{3+}+8SO_{4}^{2-}+4H^{+}$$

When the pH stabilizes (Day 11–Day 12), the pH of the leaching solutions were 8.25 (granularity 1), 7.95 (granularity 2) and 7.67 (granularity 3). The slight alkalinity highlights the strong acid neutralization capacity with the PTEs forming hydroxides or complexes after dissolution. During the storage of the waste, the PTEs dissolved and released will be reduced in their ability to migrate immediately, and in the long term the hazard potential is localized within the site, but sensitive to future change in environmental conditions [38].

## 4. Conclusions

(1) The dissolution concentration of PTEs decreases with the increase of particle size, and the dissolution intensity of each element is different. The maximum dissolution of Sb, As, Hg, Pb, Cd and Zn from 25 g samples of antimony ore waste rock were 486.128 μg, 3.219 μg, 0.207 μg, 6.104 μg, 0.025 μg and 241.152 μg, respectively, and the portions dissolved were 1.076%, 0.010%, 0.575%, 0.528%, 0.060% and 1.564%, respectively.

(2) The changes in leachate pH for the different particle sizes is consistent. The pH trends show three stages: An immediate rise, a fall and then stabilization. The main chemical reactions of the leaching system in each stage are the exchange of H^+^ with alkaline minerals and basic cations, the oxidation of reducing minerals such as pyrite to produce acid and the adsorption and precipitation of metallic elements.

(3) The waste has a strong acid neutralization capacity. The PTEs after dissolution and precipitation will form hydroxides or complexes, with lower immediate capability to migrate, but will remain available for release under changing environmental conditions. Long term impact is still likely and needs to be considered for management of the site.

## Figures and Tables

**Figure 1 ijerph-16-02355-f001:**
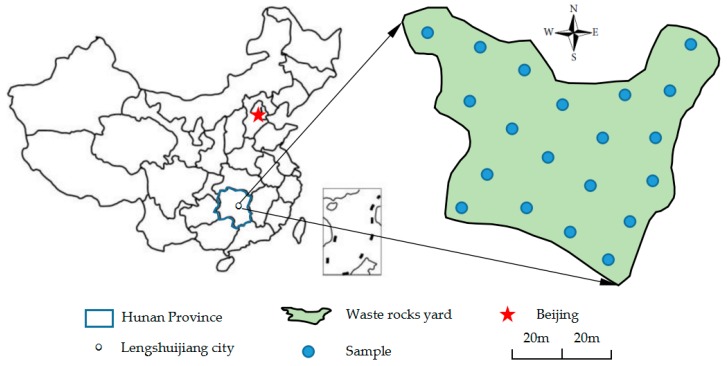
Map showing location of sampling sites.

**Figure 2 ijerph-16-02355-f002:**
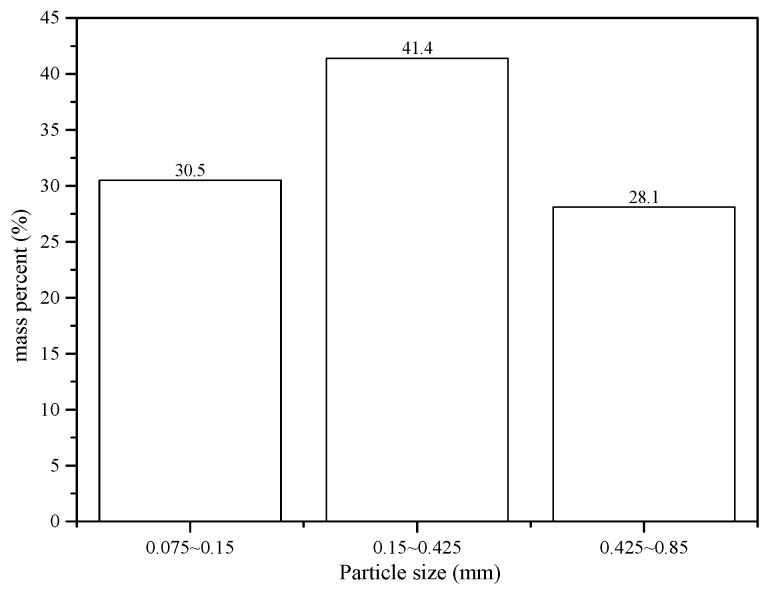
Size of distributions of antimony ore processing wastes.

**Figure 3 ijerph-16-02355-f003:**
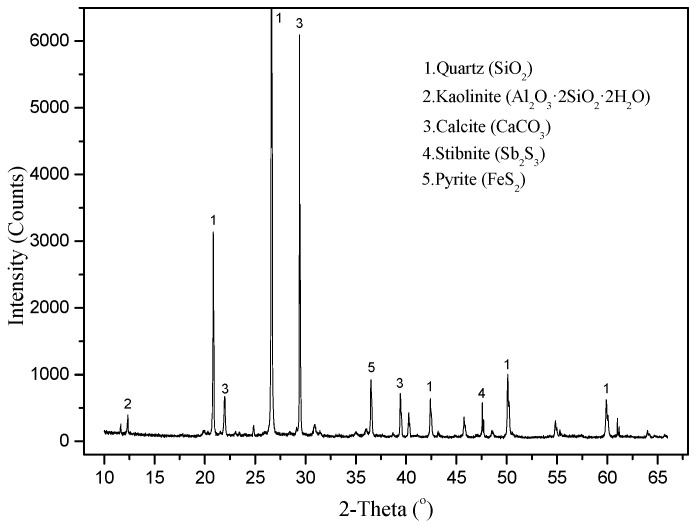
XRD analysis of typical ore waste.

**Figure 4 ijerph-16-02355-f004:**
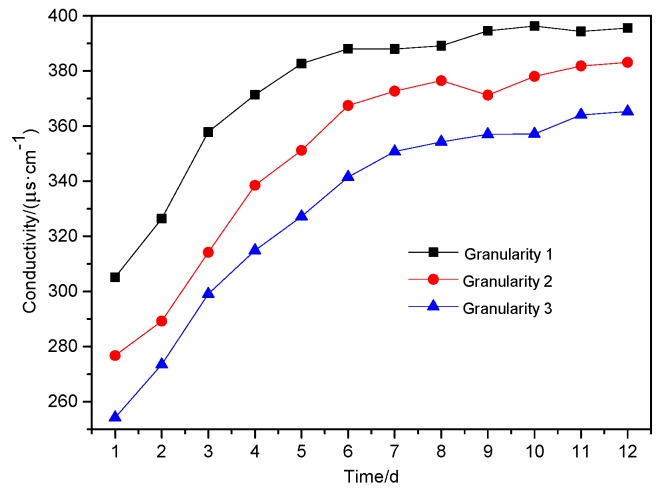
The trend in pH of leaching liquid over time with different particle size of waste.

**Figure 5 ijerph-16-02355-f005:**
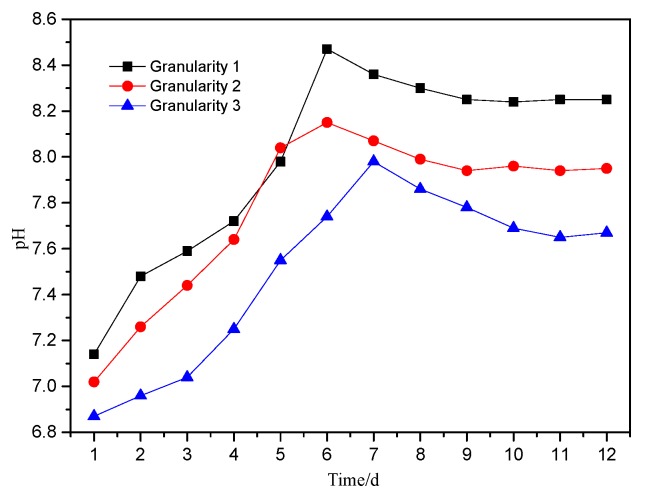
The conductivity of leaching liquid over time with different particle size of waste.

**Figure 6 ijerph-16-02355-f006:**
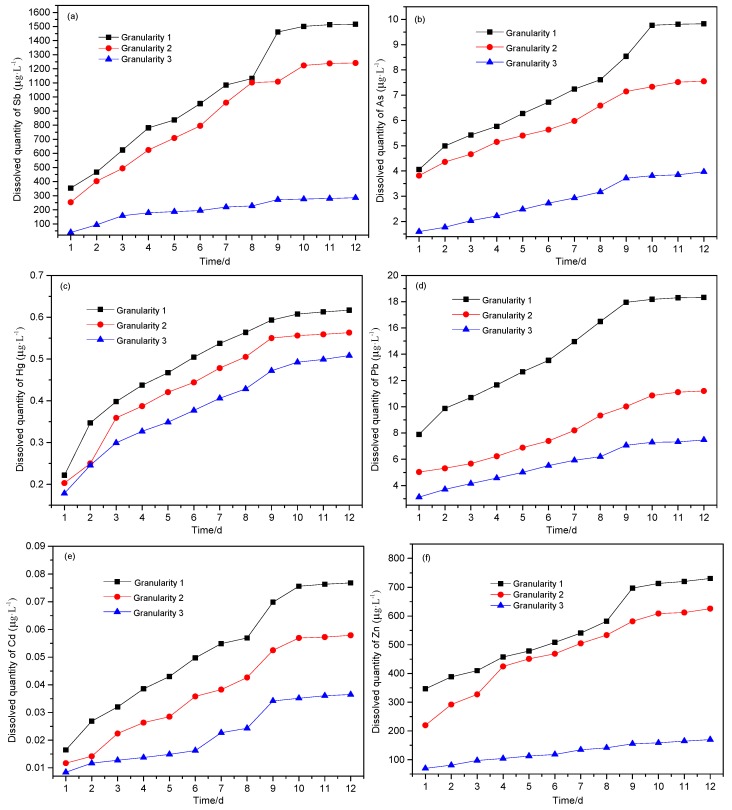
The soluble portion of PTEs over time from different solid phase particle sizes. (**a**) Sb; (**b**) As; (**c**) Hg; (**d**) Pb; (**e**) Cd; (**f**) Zn.

**Figure 7 ijerph-16-02355-f007:**
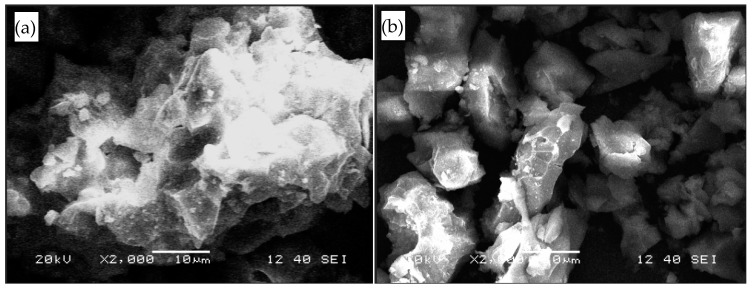
SEM images of antimony ore waste rocks at different periods (**a**) before leaching; (**b**) after leaching.

**Table 1 ijerph-16-02355-t001:** Average content and relative pollution levels of potentially toxic elements (PTEs) in waste materials.

Element	Content (mg·kg^−1^)	Background Value(mg·kg^−1^) [24]	Enrichment Quotient
Sb	1806.21	2.98	606
As	1277.64	14	91
Hg	1.44	0.09	16
Pb	46.24	27	1.7
Cd	1.67	0.079	21
Zn	616.91	95	6.5

**Table 2 ijerph-16-02355-t002:** Total leaching amount and leaching rate of heavy metals from antimony ore waste rock with different particle sizes.

	The Content of Metals (μg/kg)	Total Leaching (Granularity 1) (μg/kg)	Leached Portion (%)	Total Leaching (Granularity 2) (μg/kg)	Leached Portion (%)	Total Leaching of Granularity 3 (μg/kg)	Leached Portion (%)
Sb	45,155.25	486.128	1.076	399.363	0.884	92.626	0.205
As	31,941.00	3.219	0.010	2.523	0.008	1.295	0.004
Hg	36.00	0.207	0.575	0.188	0.522	0.168	0.467
Pb	1156.05	6.104	0.528	3.661	0.317	2.468	0.213
Cd	41.75	0.025	0.060	0.018	0.043	0.011	0.026
Zn	15,422.75	241.152	1.564	206.703	1.340	55.878	0.362

**Table 3 ijerph-16-02355-t003:** Main elements of antimony ore waste rocks surface at different periods by EDS (%, mass fraction).

	O	Si	Al	Ca	S	Sb	As	Hg	Pb	Cd	Zn	Cu	Mn	K	Fe
Before leaching	46.71	43.14	2.44	1.24	0.86	1.42	0.00	0.07	1.09	0.12	1.24	0.54	0.08	0.24	0.91
After leaching	54.03	38.82	1.23	0.85	0.69	1.25	0.61	0.03	0.78	0.06	0.93	0.05	0.07	0.02	0.58

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
