# Peer review of "The Impact of Physical Properties on the Leaching of Potentially Toxic Elements from Antimony Ore Processing Wastes"

_ijerph, 2019, doi:10.3390/ijerph16132355_

Round 1
Reviewer 1 Report
I reccommend to add more information into the Introduction section because this section is short.
Author Response
Reviewer 1
Comments and Suggestions for Authors
I reccommend to add more information into the Introduction section because this section is short.
Line 46~53
Few studies exist of the pollution caused from effects of the environment on waste rock dumps. A number of studies showed that the dissolution of Sb-bearing ore minerals is thought to contribute significantly to pollution in the local drainage zone. The kinetics of the mobilization of Sb from stibnite, Sb3O6OH and Sb2O3 under environmental conditions has been studied in detail by Biver and Shotyk[7,8]. Hu et al[23,24] studied the release kinetics and mechanisms of Sb from stibnite and Sb2O3 under the irradiation of light. Guo et al[5]. studied the leaching characteristics of antimony smelting slag in XKS Sb mine and found that smelting slag is an important source of Sb pollution in nearby farmland.
Reviewer 2 Report
The manuscript aims to investigate the heavy metals release from leaching influenced on the particle size.
There are some suggestion:
(1) Please add the map of sampling sites.
(2) Please clarify why do you choose this three particle sizes: 75-150um, 150-425um and 425-850um?
(3) Please supply this three particle sizes distribution plot measured by granularity instrument.
(4) Please give the chemical composition of three particle sizes, and then analysis the heavy metal releasing in the discussion section.
(5) how to clarify the leaching mechanism proposal in the manuscript? including of each step of chemical reaction? please give the details and discussion.
Reviewer 3 Report
The manuscript titled “Heavy Metal Static Leaching Rules affected by Different Particle Size of Antimony Ore Waste Rocks” by Saijun Zhou et al, details that antimony ore waste rocks were used as adsorbents to remove heavy metal such as Sb, As, Hg, Pb, Cd and Zn. The authors comprehensively and systematically investigated the performance of the leaching of the antimony ore waste rocks, and the results showed that antimony ore waste rock has strong capability of acid neutralization and can control the migration of heavy metals. The manuscript is clearly written and the results are well presented. Therefore, I think this manuscript could be suitable for publication after minor revision.
Specific comments:
1. The authors should provide more information that clarifies characteristic features of antimony rocks.
2. Compared with others adsorbents, what is the advantage of the antimony ore waste rocks for leaching heavy metals?
3. The heavy metal leaching performance of the antimony rocks could be affected by many factors such as the mass of absorbent, pH and temperature, which should be investigated and optimized.
Author Response
Reviewer 3
Comments and Suggestions for Authors
The manuscript titled “Heavy Metal Static Leaching Rules affected by Different Particle Size of Antimony Ore Waste Rocks” by Saijun Zhou et al, details that antimony ore waste rocks were used as adsorbents to remove heavy metal such as Sb, As, Hg, Pb, Cd and Zn. The authors comprehensively and systematically investigated the performance of the leaching of the antimony ore waste rocks, and the results showed that antimony ore waste rock has strong capability of acid neutralization and can control the migration of heavy metals. The manuscript is clearly written and the results are well presented. Therefore, I think this manuscript could be suitable for publication after minor revision.
Specific comments:
1. The authors should provide more information that clarifies characteristic features of antimony rocks.
Line 120
and also contains a small amount of stibnite and pyrite.
2. Compared with others adsorbents, what is the advantage of the antimony ore waste rocks for leaching heavy metals? revised
Line 160-162
The leaching characteristics of heavy metals in antimony ore processing wastes are consistent with previous studies [28] [29].
Line 166~167
the leaching of Sb is far below 18%[30], the reason may be related to the different pH of leaching and associated solubility .
3. The heavy metal leaching performance of the antimony rocks could be affected by many factors such as the mass of absorbent, pH and temperature, which should be investigated and optimized. revised
This paper is to study the leaching release of waste rock in XKS antimony mine under local simulated rainfall conditions. The mass of adsorbent and temperature, which should be investigated in the next paper.

Round 2
Reviewer 2 Report
The manuscript has been revised a few points according to the comments.
I did not read the revisions about the following important
suggestions proposed in the first review.
(1) Please clarify why do you choose this three particle sizes: 75-150um, 150-425um and 425-850um?
The authors response "Refer to the other people's research", so please give the details in the manuscript. It is important to your research.
(2) Please supply this three particle sizes distribution plot measured by granularity instrument.
The authors response "The samples of different particle size are separated by circular hole sieve", so please add the partice size distribution figures in manuscript after using instruments meansure in order to display the accuracy of size properties.
